# Coronavirus: Public Arabic Twitter Dataset

## Abstract

The COVID-19 pandemic spread of the coronavirus across the globe has affected our lives on many different levels. The world we knew before the spread of the virus has become another one. Every country has taken preventive measures, including social distancing, travel restrictions, and curfew, to control the spread of the disease. With these measures implemented, people have shifted to social media platforms in the online sphere, such as Twitter, to maintain connections. In this paper, we describe a coronavirus data set of Arabic tweets collected from January 1, 2020, primarily from hashtags populated from Saudi Arabia. This data set is available to the research community to glean a better understanding of the societal, economical, and political effects of the outbreak and to help policy makers make better decisions for fighting this epidemic.

## 1 Introduction

After the wide spread of the coronavirus (COVID-19) began, the World Health Organization declared it a pandemic on January 30, 2020 (World Health Organization et al., 2020). The first case was reported as originating in the city of Wuhan, China, where the government had to quarantine the whole city to overcome the quick spread of the disease. However, with globalization and the way the modern world functions, the pandemic has affected 213 countries, with more than one and a half million confirmed cases to date (World Health Organization et al., 2020).

This spread has led governments around the globe to start implementing crisis management plans and pandemic control strategies. Although governments and public health authorities may implement prevention measures and control policies, the public plays a vital role in following these measures to contain the spread of the disease.

The most important measures used to combat the spread of the virus are limiting physical contact between people and reducing the time people spend next to one another. People now rely more on the internet and online platforms to continue their social interactions. One of the most widely used social media platforms is Twitter, popular for its accessibility and ease of information sharing.

In this work, we focus on Arabic online conversation because Arabic is ranked fourth among the top 10 languages used on the web [1]. The main focus in the data set collected was on hashtags used in Saudi Arabia, although they might be used in Arabic-speaking countries outside of Saudi Arabia. Saudi Arabia is among the countries with the highest number of Twitter users among its online population (Clement, 2020; Puri-Mirza, 2019). Moreover, Saudi Arabia produces 40% of all tweets in the Arab world (Mourtada and Salem, 2014).

The data set shared is divided into conversations discussing the precautionary measures governments have applied, conversations showing social solidarity, and conversations supporting decisions governments have taken. The data set also contains data from three Saudi official accounts. The total number of tweets collected so far is 3.8 million.

Policy and decision makers can use the described data set to understand people's engagement in social media and to track the spread of misinformation.

In the following sections we describe data collection, data set statistics, and information about

---

[1] https://www.internetworldstats.com/stats7.htm

Table 1: Hashtags discussing precautionary measures applied by governments.

| Hashtags | English Translation | Number of Tweets |
| --- | --- | --- |
| إيقاف _صلاة _الجماعة | Stopping congregational prayer | 7502 |
| إغلاق _الحدائق | Gardens closure | 180 |
| صلوا _في _رحالكم | Pray in your travel | 37997 |
| ايقاف _الصلاة _بالمسجد | Stop praying in the mosque | 2953 |
| ايقاف _صلاة _الجمعة _والجماعة | Stopping Friday and group prayers | 79 |
| إغلاق _محلات _الحلاقة | barber shops closure | 686 |
| اغلاق _المقاهي | Cafes closure | 30 |
| اغلاق _الصالونات | Salons closure | 13466 |
| ايقاف _الدوري | Stopping football league | 18923 |
| تعليق _النشاط _الرياضي | Sports suspension | 2824 |
| تعليق _الرحلات _الدولية | International flights suspended | 102 |
| تعليق _الرحلات _الداخلية | Internal flights suspended | 94 |
| تعليق _العمل | Work suspension | 35170 |
| تعليق _الدراسة | School Suspension | 127064 |
| اغلاق _النوادي _الرياضية | Gym closure | 1625 |
| --- | Close the malls in Saudi Arabia | 7614 |
| إغلاق _المجمعات _التجارية | Close the commercial complexes | 239 |
| تعليق _القطاع _الخاص | private sector suspension | 18962 |
| منع _التجول | Curfew | 199925 |
| منع _التنقل _بين _المناطق | Prevent movement between regions | 16908 |

Table 2: Hashtags showing social solidarity.

| Hashtags | English Translation | Number of Tweets |
| --- | --- | --- |
| المسافة _ما _تفرقنا | distance does not separates us | 35,498 |
| نتفرق _لصحتنا | Lets Separate for our health | 27,612 |
| احنا _قدها | We can do it | 8,889 |
| سحابة _وتعدي | A cloud and it will pass | 18,800 |
| يارب _ارفع _عنا _البلاء | Oh Lord, raise us from calamity | 72,401 |
| مبادرة _أنتم _أبطال | You are a champion initiative | 1,440 |
| تجارنا _فيهم _الخير | Our merchants are good | 27,714 |
| ابطال _الصحة _بكم _نفخر | Health champions we are proud of you | 12,720 |
| البيض _متوفر | Eggs_available | 14,694 |

Table 3: Hashtags supporting decisions taken governments.

| Hashtags | English Translation | Number of Tweets |
| --- | --- | --- |
| اسبوعين _فقط | Only two weeks | 29,355 |
| تمرن _ببيتك | Exercise at home | 2,538 |
| بيتك _ناديك | Your home is your gym | 5,425 |
| بتمرن _بالبيت | Exercise at home | 40,196 |
| اجلس _بالبيت | Sit at home | 69,065 |
| قهوتك _في _بيتك | Your coffee in your home | 696 |
| نبها _صفر | We want it zero | 94,627 |
| المملكة _تستاهل _اكثر | The kingdom deserves more | 51,914 |
| كلنا _مسؤول _عن _الوطن | We are all responsible for the country | 5,479 |
| ليه _ياوطن | To him, my homeland | 17,572 |
| اكثر _شي _سويته _بالحجر _المنزلي | The most thing you did in the Quarantine | 30,911 |
| الحجر _المنزلي _واجب _وطني | Quarantine is a national duty | 55,942 |
| حظر _كامل | Curfew | 82,349 |
| كلنا _بالبيت _لاجل _السعوديه | We are all home for Saudi Arabia | 34,340 |
| حظر _التجول _الكامل | Curfew | 37,862 |
| حجر _كامل | Curfew | 15,579 |
| قاعد _بالبيت | Staying home | 64,355 |
| خلك _بالبيت | Stay home | 177,619 |
| خلك _في _البيت | Stay home | 220,759 |
| اثبروا _ببيوتكم | Stay home | 30,683 |
| حظر _التجول _في _السعودية | Curfew in Saudi Arabia | 36,292 |
| وضعنا _مع _الحجر | Our situation with Quarantine | 21,059 |
| استراحتي _في _بيتي | My rest in my home | 5,479 |
| نشاطي _في _منزلي | My activity in my home | 22,750 |
| فعاليات _الحجر _الصحي | Quarantine activities | 24,239 |
| الحجر _المنزلي | Quarantine | 362,132 |
| كلنا _في _البيت _لاجل _السعوديه | We are all at home for Saudi Arabia | 77,021 |
| الزم _بيتك _حمايه _لك _ولمجتمعك | stay home for you and your community safety | 31,584 |
| الحجر _الصحي | Quarantine | 141,311 |
| اعزل _نفسك | Isolate yourself | 25,344 |

how to access the data set.

## 2 Data Collection and Description

The data collection started by identifying a list of trending hashtags and key words mostly used by the public. We used *Crimson Hexagon*, [2] , which is a social media analytic platform that provides paid data stream access. This tool allowed us to obtain tweets and retweets discussing the epidemic in Arabic. We collected data starting from January 1, 2020, until April 10, 2020, collecting 3.8 million tweets until that date. More data will be collected as the project continues.

To capture conversations related to the epidemic and people's reactions toward it, we continuously observed trending topics and hashtags. Around 70 keywords and hashtags were selected; these were later categorized based on how they oriented the conversations because this is the main purpose of hashtags. Table 1 lists hashtags that mainly discuss precautionary measures governments have applied. These include discussions of curfew, business closures, and travel restrictions. Table 2 lists hashtags that show some kind of social solidarity within the community after applying such prevention measures as social distancing. This category includes such hashtags as "distance does not separate us."

Table 3 lists hashtags that show support for the decisions and prevention measures governments have taken. This group includes hashtags encouraging people to stay home, exercise at home, or enjoy

their time while in quarantine.

Table 4 lists hashtags populated by Saudi governmental Twitter accounts. These hashtags urge the community to be responsible about decreasing the number of cases by following prevention measures, reassure the community about the availability of products, and answer common questions about COVID-19. The table shows the list of hashtags accompanied by the governmental account from where they started.

One of the main ways to overcome misinformation is to take information from known, reputable sources. In social media, the most reliable sources are governmental sources. Table 5 below lists the Saudi Arabia Ministry of Health accounts and the number of tweets collected for each account.

Preliminary statistics are given in Table 6. The table shows the number of tweets, the number of retweets, the number of unique users, and the num-

---

[2]https://www.crimsonhexagon.com/

Table 4: Hashtags populated by Saudi governmental accounts.

| Hashtags | English translation | Account | Number of Tweets |
|---|---|---|---|
| الوقاية _من _كورونا | Corona Prevention | @SaudiMOH | 293,123 |
| كنا _مسؤول | We are all responsible | @spokesman_moh | 596,288 |
| عش _بصحة | Live healthily | LiveWellMOH | 28,541 |
| أسئلة _كورونا | Corona's Questions | @SaudiMOH | 15,455 |
| أبطال _الصحة | Health Heroes | KSAMOFA | 155,770 |
| أبطال _المجتمع | Community Heroes | @SaudiMOH | 16,641 |
| المنتجات _متوفرة | Products available | @MCgovSA | 22,992 |
| الخدمات _مستمرة | Services continuous | @MCgovSA | 4,501 |
| متر _ونص | One and a half meters | @SaudiMOH | 8,616 |
| شكراً _أبطال _التعليم | Thanks Education heroes | @moe_gov_sa | 18,227 |
| الدراسة _مستمرة | Schools are continuing | @moe_gov_sa | 152,901 |

Table 5: Saudi Arabia ministry of health accounts.

| Account name | Number of Tweets |
|---|---|
| @spokesman_moh | 66 |
| @LiveWellMOH | 637 |
| @SaudiMOH | 897 |

Table 6: Descriptive statistics of the dataset.

| | |
|---|---|
| Number of original tweets | 707,829 |
| Number of retweets | 3,093,026 |
| Number of unique users | 1,000,244 |
| Number of unique users who wrote original tweets | 323,876 |

ber of unique users who wrote original tweets. These numbers will expand as we continue to monitor Twitter for additional hashtags and keywords to add to our tracking list.

## 3 Dataset Access

The data set is accessible through GitHub at this address:

To comply with Twitter's Terms and Conditions[3], we are unable to distribute the text of the collected data set. For that, only tweet IDs can be released and then used to retrieve the full tweet object. To do so, some tools have been developed to make the process easier; Hydrator[4] is one of these options.

## Acknowledgements

The author gratefully thanks Afnan abdullah Aloqaily and Djedaini abdelhak for their help in data collection.

---

[3]https://developer.twitter.com/en/ developer-terms/agreement-and-policy

[4]https://github.com/DocNow/hydrator

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
