# OpenReview forum: "Coronavirus: Public Arabic Twitter Data Set"
_EMNLP/2020/Workshop/NLP-COVID — Submitted to NLP-COVID19-EMNLP_

### Official Review · AnonReviewer3 · 2020-09-07
**Good resource but remains shallow**

**Rating:** 4
**Confidence:** 5

**Review:**

I think this is a good resource (esp. given it is in Arabic and thereby focuses on other-than-English data).
However, the paper in its current form falls short in providing any deeper look at the dataset. There are also a few relevant references missing.

Detailed points:

- line 45-47: consider updating the infection numbers or state the date when the numbers were accurate
- line 69-71: needs a reference
- line 89: which government accounts?
- line 124-130: what are the limitations of that tool? How is the sampling done? How do you know which part of the Tweet population you are getting? Some references you’d want to look at here are [1], [2] and [3] (see below:
- line 135-137: how were they categorised? By whom? What was the inter-rather agreement?
- line 193-194: needs reference
- line 233: the link is not stated so I cannot access the data

Overall: this paper would do well to also include a section on previous work - esp that on COVID-19 and NLP, many of which has been published at the COVID-19 ACL workshop. Relevant datasets, studies and findings should be discussed.

I would also want to see analyses on the dataset. E.g. which topics are discussed in the tweets (using LDA, for example)? What can we learn from the data?

A key limitation is further see is that it focuses on Twitter without discussing the limitations that follow.

References:

[1] https://arxiv.org/pdf/1306.5204.pdf%3B
[2] https://science.sciencemag.org/content/346/6213/1063.summary
[3] https://arxiv.org/pdf/1403.7400.pdf

---

### Official Review · AnonReviewer2 · 2020-09-07
**Interesting resource but requires more work**

**Rating:** 4
**Confidence:** 4

**Review:**

#### General review

The paper presents a dataset of over 3 million tweets in Arabic collected during the initial months of the COVID-19 pandemic. Authors selected relevant hashtags related to public policies, measures, solidarity messages, etc. The paper presents the main statistics of the corpus and briefly discusses the data collection methodology.

I commend the authors for their effort in this endeavour. The creation of new linguistic resources is always a time-consuming process, but it is extremely valuable for research in NLP. Furthermore, resources in languages other than English are necessary to diversify our methods and technologies and ensure they are scalable to these scenarios. In this respect, I consider the work relevant.

However, my main concern is that I think the work is still very initial to be considered for presentation in EMNLP. At this moment, the authors have "only" collected the data (which I agree is an important part of the research), but it remains to be seen how this data can be used to successfully answer some interesting questions in NLP. Furthermore, the description of the dataset and data collection methodology is not sufficiently detailed in order to make this process as transparent and reproducible as possible.

Hence, unfortunately, I have to recommend that this work is rejected at this moment, while at the same time I recommend the authors to continue their research and submit an updated and extended version in future venues.

#### Some questions I consider could help the authors improve their work:

- How was the selection of hashtags and categorization? Was it the work of a single person, or was some sort of committee designed, and in that case, how was disagreement dealt with? If the selection was performed by a single person, then it is harder to argue that the hashtags and categories are objectively meaningful, but at least is important to disclose this information.

- Can authors detail which (if any) strategies were used for filtering or identifying unwanted messages (e.g., hate speech) or irrelevant messages (e.g., I have anecdotally seen that replies to an official source often contain a large portion of messages which are plain spam, completely unrelated to the original message).

- Is there some classic NLP task that the authors can show, at least with simple baselines, can be aided by or performed in this dataset? For example, apply sentiment analysis to the tweets and estimate a percentage of agreement/disagreement between retweets and original messages.

- Can authors provide additional statistics of the dataset, for example, timelines showing the prevalence of different hashtags correlated with major events in the Arab world (e.g., curfew application). Similarly, an analysis of the most common terms used in different categories. The idea is to give the reader a view as deep as possible into the content of the dataset such that they can judge if it will be useful for some specific task.

- Regarding collected retweets, are these plain retweets, or retweets with comments, or both? I think plain retweets, even if valuable to understand how a tweet flows through the network, are less relevant than retweets with comments which can also be used to estimate stance or opinion w.r.t. the original message.

#### Minor suggestions:

- Line 43: Update to more recent statistics and include the specific date to which "to date" refers.
- Line 233: I couldn't find the dataset repository URL in the document.

Once again, I commend the authors for their hard work so far and recommend they continue working on this very important line of research.

---

### Official Review · AnonReviewer4 · 2020-09-27
**Worthwhile effort, but needs strengthening**

**Rating:** 4
**Confidence:** 4

**Review:**

# [REVIEW]  Coronavirus:  Public Arabic Twitter Dataset

EMNLP COVID-19 WORKSHOP — 26th Sep 2020

## SUMMARY

This short paper/abstract describes the development of a corpus of Arabic COVID-19 related tweets from Saudi Arabia.   Saudi has a relatively high proportion of Twitter users and produces around 40% of all Arabic tweets.  Data was collected between Jan 1st 2020 and Apr 10th 2020, and consisted of 3.8 million tweets selected based on Arabic COVID-related hashtags.

The development of this corpus is a useful endeavour, but it is not clear to this reviewer that in and of itself it warrants publication.  The case for publication would be stronger if the corpus was manually annotated and/or there was a more extensive analysis of corpus characteristics.

## COMMENTS

1.  Generally, the language is a bit hard to follow.
2.  It is not clear how you identified the hashtags.